# Molecular Characterization of a Novel Strain of Fusarium graminearum Virus 1 Infecting *Fusarium graminearum*

**DOI:** 10.3390/v12030357

**Published:** 2020-03-24

**Authors:** Lihang Zhang, Xiaoguang Chen, Pallab Bhattacharjee, Yue Shi, Lihua Guo, Shuangchao Wang

**Affiliations:** 1State Key Laboratory for Biology of Plant Diseases and Insect Pests, Institute of Plant Protection, Chinese Academy of Agricultural Sciences, Beijing 100193, China; LHzhang0911@163.com (L.Z.); xiaoguang.chen@pflern.uni-hannover.de (X.C.); pallabsau19@gmail.com (P.B.); guolihua72@yahoo.com (L.G.); 2Agricultural Trade Promotion Center, Ministry of Agriculture and Rural Affairs, Beijing 100125, China; shiyue8569@163.com

**Keywords:** mycovirus, *Fusarium graminearum*, FgV1-ch, latent infection

## Abstract

Fungal viruses (mycoviruses) have attracted more attention for their possible hypovirulence (attenuation of fungal virulence) trait, which may be developed as a biocontrol agent of plant pathogenic fungi. However, most discovered mycoviruses are asymptomatic in their hosts. In most cases, mycovirus hypovirulent factors have not been explored clearly. In this study, we characterized a ssRNA mycovirus in *Fusarium graminearum* strain HB56-9. The complete nucleotide genome was obtained by combining random sequencing and rapid amplification of cDNA ends (RACE). The full genome was 6621-nucleotides long, excluding the poly(A) tail. The mycovirus was quite interesting because it shared 95.91% nucleotide identities with previously reported Fusarium graminearum virus 1 strain DK21 (FgV1-DK21), while the colony morphology of their fungal hosts on PDA plates were very different. The novel virus was named Fusarium graminearum virus 1 Chinese isolate (FgV1-ch). Like FgV1-DK21, FgV1-ch also contains four putative open reading frames (ORFs), including one long and three short ORFs. A phylogenetic analysis indicated that FgV1-ch is clustered into a proposed family Fusariviridae. FgV1-ch, unlike FgV1-DK21, had mild or no effects on host mycelial growth, spore production and virulence. The nucleotide differences between FgV1-ch and FgV1-DK21 will help to elucidate the hypovirulence determinants during mycovirus–host interaction.

## 1. Introduction

Fungal viruses or mycoviruses are widespread in fungi and a large number of viruses that infect phytopathogenic fungi have been found [1]. According to the genomic type, mycoviruses can be classified as single-stranded DNA (ssDNA) viruses, double-stranded RNA (dsRNA) viruses and single-stranded RNA (positive-sense, negative-sense, and reverse transcribing) viruses [2]. The International Committee on Taxonomy of Viruses (ICTV) grouped fungal viruses with positive-sense ssRNA into six families: *Alphaflexiviridae, Barnaviridae, Deltaflexiviridae, Gammaflexiviridae, Hypoviridae, Narnaviridae and Endornaviridae* [3]. In host cells, most discovered ssRNA mycoviruses exist in the form of dsRNA genome.

Fusarium head blight (FHB), mainly caused by *Fusarium graminearum sensu stricto,* is a devastating disease that causes extensive yield losses in most of the world’s major wheat-growing areas [4]. Trichothecenes and Zearalenone, which were produced by *F. graminearum*, cause great economic loss by humans and livestock health risk. In China, FHB outbreaks have occurred much more frequently in recent years [5,6]. Extensive application of chemical fungicide has caused serious environmental pollution and fungal tolerance problems [7]. Nowadays, mycoviruses are being developed as potential biological control agents of *Fusarium*, as some characterized mycoviruses can attenuate fungal growth and pathogenicity.

Isolated mycoviruses from *Fusarium* have attracted attention in wheat-, corn- and vegetable-growing countries, especially in China and South Korea [8]. Many *Fusarium* viruses have been discovered and characterized, such as Fusarium graminearum virus 1, 2, 3 and 4 (hereafter referred to as FgV1-DK21, FgV2, FgV3, and FgV4, respectively) and Fusarium graminearum hypovirus 1 (FgVH1). Among them, the most in-depth studied mycovirus is FgV1-DK21, which can severely reduce growth and pathogenicity in its host [9]. Translational and proteomic changes responding to FgV1-DK21 infection have been elucidated. Genes associated with protein synthesis, transcription and signal transduction, metabolic pathways and transport systems were differentially expressed in FgV1-DK21-infected strain. Proteins including sporulation-specific protein SPS2, triose phosphate isomerase, nucleoside diphosphate kinase, woronin body and malate dehydrogenase were regulated by FgV1-DK21 infection [10]. Moreover, the FgV1-DK21 ORF2-encoded protein (pORF2) can bind to the promoters of host RNA-silencing critical genes, *FgDICER2* and *FgAGO1*, which may facilitate FgV1-DK21 infection [11]. As for most other characterized *Fusarium* viruses, virus–host interactions were not studied in depth.

In this paper, we characterized a new strain of FgV1 in *Fusarium* from China, which was named FgV1-ch. FgV1-ch shares a high genome sequence identity with FgV1-DK21. However, FgV1-DK21 is hypovirulent whereas FgV1-ch is asymptomatic. The minor nucleotide changes probably contain the code of dramatic phenotype differences between FgV1-DK21 and FgV1-ch.

## 2. Materials and Methods

### 2.1. Fungal Isolates and Culture Conditions

*Fusarium* spp. strains were collected from wheat ear with typical FHB symptoms from the field in main wheat-producing areas with different environment conditions in China. All the collected strains were cultured and sub-cultured at 25 °C on PDA (potato dextrose agar) plate in the dark. The strain HB56-9 was obtained by single spore isolation and pure culturing. Mycelial plugs and asexual spores were stored in 25% glycerol at -80 °C. The species of *Fusarium* spp. strains were identified by sequencing of the translation elongation factor-1alpha (*EF-1α*) gene. Genomic DNAs were extracted using a DNA extraction kit (Transgen, Beijing, China) and then amplified with *EF-1α* primer pair, ef1: (5′-ATGGGTAAGGA(A/G)GACAAGAC-3′); ef2: (5′-GGA(A/G)GTACCAGT(G/C)ATCATGTT-3′) [12].

### 2.2. dsRNA Extraction and Enzymatic Digestions

Mycelial plugs of *Fusarium* strains were cultured on PDA plates overlaid with cellophane membranes for 5 days at 25 °C in the dark. Then, mycelial masses were collected. Viral dsRNA was extracted by cellulose (Sigma, Dorset, England) chromatography [13]. DNase Ι and S1 nuclease (TaKaRa, Shiga, Japan) were used to eliminate DNA and ssRNA contamination according to the manufacturer’s instructions. The dsRNA samples were electrophoresed on 1 percent agarose gels (Coolaber, Beijing, China) stained with ethidium bromide and visualized under 350-nm UV illumination.

### 2.3. Complementary DNA (cDNA) Cloning and Sequence Analysis

M-MLV Reverse Transcriptase (Promega, Madison, WI, USA) and PrimeSTAR^®^ HS DNA polymerase (TaKaRa, Japan) were used for cDNA synthesis and PCR in a Thermal Cycler (Bio-Rad, Richmond, VA, USA) based on the protocol, respectively. The RT-PCR products were purified using a gel extraction kit (Sigma, Deisenhofen, Germany) and were ligated to PMD18-T vector and transformed into *Escherichia coli* strain DH5α (TaKaRa, Japan) for sequencing. Based on the obtained sequences, specific primers were designed to amplify the gap sequences [14]. The 3′ RNA ligase-mediated RACE method was used to determine the terminal sequences of the dsRNA as described previously [15,16].

### 2.4. Virus Transmission

The hyphal anastomosis method was used to transmit the virus from an infected fungal strain into a virus-free *F. graminearum* strain PH-1, incorporating a hygromycin-resistance gene (hereafter referred to as PH-1). During hyphal anastomosis, PH-1 was used as the recipient and the virus-infected HB56-9 strain was used as the donor. The donor and recipient strains were inoculated together on the same PDA plate and incubated at 25 °C for 5 days. Mycelial plugs were taken from the growth side of PH-1 and transferred on a PDA plate containing 200μg/mL hygromycin B (Coolaber, China). We collected the mycelial plugs that grow on the PDA plate containing hygromycin-B and then isolated single spores. The spores were examined with dsRNA extraction and RT-PCR process. The fungal strain derived from PH-1 infected with the virus was named PH-1-V.

### 2.5. Impact of the Virus on Host Biological Properties

Colonial diameters were measured after culturing mycelial plugs on PDA plates for 96 h at 25 °C in the dark. Conidial production was determined with a hemocytometer after culturing 5 days in carboxymethyl cellulose (CMC) medium at 25 °C in the dark. For exploring the virulence of FgV1-ch, the virulence assay was conducted on wheat heads as described previously [17]. A single spikelet of the wheat head was inoculated with asexual spores at a concentration of 3 × 10^5^/mL, and then infected spikelets were counted as described previously [18]. All the experiments were repeated three times independently. Statistical analyses were performed using the SAS 8.0 software paired t-test at a significant level of *p* < 0.05.

### 2.6. Sequence and Phylogenetic Analysis

The obtained sequences and translation of ORFs from nucleotides were assembled and analyzed, using DNAMAN version 8 software (Lynnon Biosoft, Quebec, Canada). Sequences were aligned with the ClustalW program. Sequence similarity searches were performed in NCBI BLAST program (Blastn, Blastp). The construction of phylogenetic tree was carried out with the neighbor-joining method in MEGA 6 program [19]. The potential RNA secondary structure was analyzed using the UNAFold Web Server (http://unafold.rna.albany.edu).

## 3. Results

### 3.1. dsRNA in F. graminearum Strain HB56-9 and Colony Morphology

About 200 *Fusarium* isolates were recovered from wheat samples collected from individual croplands in seven provinces, which are the main wheat production areas in China. The dsRNA molecule is an indicator of RNA mycovirus infection and dsRNAs were purified from these strains to confirm which strain was infected by mycovirus(es). As shown in Figure 1A, one dsRNA fragment with a size of approximately 8 kb was recovered from strain HB56-9 and collected from Hebei province of China. The purified dsRNA was digested with DNase Ι and S1 nuclease to eliminate DNA and ssRNA contamination. Based on the BlastN results of the *EF-1α* gene sequence, the HB56-9 strain was identified as *F. graminearum*.

To obtain a partial sequence of this dsRNA element, cDNAs of the gel-purified dsRNA were synthesized with random primers-dN6 (5′-GACGTCCAGATCGCGAATTCNNNNNN-3′). The resulting cDNAs were amplified using a single specific primer (5′-GACGTCCAGATCGCGAATTC-3′), and after amplification, they were cloned and sequenced. The analysis of the obtained sequences with blast program revealed significant sequence similarities with FgV1-DK21 (data not shown), which was also characterized in *F. graminearum*. The FgV1-like mycovirus is intriguing because the mycovirus-infected *F. graminearum* strain HB56-9 showed a normal cultural morphology on the PDA plate (Figure 1B), while the FgV1-infected *F. graminearum* DK21 strain exhibited a quite different phenotype, including obviously reduced mycelial growth and increased pigmentation.

### 3.2. Full Nucleotide Sequence of dsRNA

To obtain the full nucleotide sequence of FgV1-like mycovirus, the obtained random sequences were assembled using DNAMAN and then, the gap between contigs was filled by PCR with specific primers. The 5′ and 3′-terminal sequences were determined by RLM-RACE. Three to fourteen independent clones were sequenced for every single base. After assembly, the complete FgV1-like mycovirus genome was 6621 nucleotides, excluding the 3′ poly A tail. According to NCBI BlastN results, the complete sequence of the mycovirus showed 95.91% identity to FgV1-DK21. In view of the high similarity of the dsRNA obtained in this study to FgV1-DK21 isolated from Korea, the mycovirus from *F. graminearum* strain HB56-9 isolated from China was considered a new isolate of FgV1 and tentatively named Fusarium graminearum virus 1-ch (FgV1-ch). The complete nucleotide sequence of FgV1-ch was deposited under accession number MT024571 in GenBank.

### 3.3. FgV1-ch Genome Organization and Sequence Comparisions

The genome organization of FgV1-ch was outlined based on a sequence analysis, as shown in Figure 2A. Four putative open reading frames (ORFs), ORF1-4, in the nucleotide sequence were revealed by the sequence analysis, two of which are long (ORF1 and ORF4) and two are short (ORF2 and ORF3). The 5′ and 3′ untranslated regions (UTRs) were 53 and 46 nts, respectively. ORF1, beginning at 54 nt and terminating at 4706 nt, encodes the largest putative protein with a predicted molecular mass of 174 kDa. Based on the NCBI domain research results, RNA-dependent RNA polymerase (RdRp) and Helicase (Hel) domains were detected within ORF1. A Blastp search using the predicted amino acid sequences of ORF 4, ORF2 and ORF3 showed that ORF4 has a Chromosome Segregation protein (provisional) (CS) domain and no conserved domain was predicted in ORF2 or ORF3. FgV1-ch had similar genome organization with FgV1-DK21, which also contains four ORFs at the same nucleotide positions. Full genome sequence alignment of FgV1-ch and FgV1-DK21 revealed the distribution of nucleotide differences along the whole genome of FgV1-ch (Figure 2B).

In detail, the nucleotide identities of 5′-UTR, ORF1-4, 3′-UTR between FgV1-ch and FgV1-DK21 were 100%, 95.44%, 96.13%, 98.77%, 97.21% and 95.65%, respectively. Pairwise amino acids between putative proteins encoded by ORF 1-4 of FgV1-ch and proteins encoded by FgV1-DK21 were 99.03%, 96.75%, 98.11% and 98.60%, respectively. The putative amino acid sequence of RdRps of FgV1-ch and ten selected members of the family Fusariviridae were aligned using the ClustalW program (Figure 3A). Multiple alignments revealed eight conserved motifs, including the highly conserved GDD motif, typical of RdRps in motif VI in putative RdRp of FgV1-ch. A phylogenetic analysis was conducted based on alignments of amino acid RdRp sequences. Based on this analysis, FgV1-ch was clustered with FgV1-DK21 in a clade within the proposed family Fusariviridae (positive-sense ssRNA viruse family) (Figure 3B).

### 3.4. Impact of FgV1-ch on Host Biological Properties

To determine the biological effect of FgV1-ch on *F. graminearum*, FgV1-ch was transmitted into *F. graminearum* strain PH-1, which had a hygromycin B resistance gene. The FgV1-ch-infected PH-1 strain, which is syngenic to PH-1, was named PH-1-V in this paper. As shown in Figure 4A, the colony morphology of PH-1-V and PH-1 shows minor differences and the growth rate of PH-1-V was 15% slower than that of PH-1 on PDA plate (Figure 4B). Meanwhile, compared with PH-1, PH-1-V produced 10% less asexual spores in CMC medium (Figure 4C). In the virulence assay, PH-1-V hyphae spread to close spikelets from inoculation point at the same speed as PH-1. After 15 days post inoculation, the number of PH-1-V infected spikelets per wheat head was not significantly different from PH-1 (11 infected spikelets on average), which signifies no difference in virulence between PH-1-V and PH-1 (Figure 4D). As mentioned above, there are mild morphological differences between PH-1-V and PH-1, but there is no difference in virulence between two strains.

## 4. Discussion

In some cases, the growth and virulence of fungi can be reduced by mycovirus infection. The resources and understanding hypovirulence mechanism of mycoviruses are of great importance for the potential biological control of fungal diseases [20]. To date, mycoviruses have been isolated from various fungi, and have had different effects on the virulence and other physiological properties of host fungi. For example, a 6-kbp dsRNA virus confers enhanced virulence to the host fungus, *Nectria radicicola*; this phenomenon is called “hypervirulence” [21]. In contrast, the term “hypovirulence” is well exemplified as reduced fungal virulence [22]. Hypovirulence is usually associated with other phenotypic changes, such as increased pigmentation, reduced sporulation and defects in growth. As a long-established system, Cryphonectria hypovirus 1/*Cryphonectria parasitica* is a good tool for the study of virus–fungi interactions [23]. However, most mycoviruses are not well studied for their hypervirulence or hypovirulence effects.

In this study, we identified a 6621 bp virus associated with *F. graminearum* that we tentatively named FgV1-ch. FgV1-ch has a similar genomic organization and a high sequence identity to FgV1-DK21. FgV1-ch and FgV1-DK21 were isolated from close geographical areas (China and Korea) and in phylogenetic analyses, they were clustered together with strong bootstrap support, suggesting that they may have a close evolutionary relationship. However, their effect on the host phenotype are quite different. *F. graminearum* infected by FgV1-DK21 showed a quite abnormal colony morphology and a significant virulence decline (hypovirulence). Although host mycelial growth rate and conidial production were slightly reduced, host virulence was not affected significantly (no hypovirulence) by FgV1-ch infection. The limited nucleotide differences between FgV1-ch and FgV1-DK21 may play a crucial role in symptom induction.

Based on sequence alignment, RdRp protein is highly conserved between FgV1-ch and FgV1-DK21 with only 15 amino acid differences. The other putative proteins encoded by FgV1-ch and FgV1-DK21 also share high sequence identities. The putative proteins encode by ORF2, ORF3 and ORF4 of FgV1-ch showed five, one and six amino acid differences with proteins of FgV1-DK21, respectively. It is worth noting that among the six different amino acids of the putative protein encoded by ORF4 of FgV1-ch compared to FgV1-DK21, there are several serine and threonine changes, which may be phosphorylated easily and lead to functional changes [24,25]. As reported, SWI6, which is involved in the regulation of meiotic initiation, plays an important role in FgV1-DK21-induced phenotype alteration [26]. It is possible that mutations in ORF4 of FgV1-ch, containing chromosome segregation (provisional) domain, may influence the coupling effects with SWI6 in host cell.

It has been demonstrated that FgHal2 deletion mutant of *F. graminearum* showed a similar abnormal morphology to the FgV1-DK21-infected strain. Moreover, the *FgHal2* expression level was decreased by FgV1-DK21 infection [27]. Therefore, it will be interesting to check whether sequence differences between FgV1-ch and FgV1-DK21 could lead to a different *FgHal2* expression. As we know, RNA silencing plays an important role in virus infection defense. It has been demonstrated that FgV1-DK21-encoded pORF2 could interfere with the induction of RNA-silencing-related genes *FgDICER-2* and *FgAGO*-1 in a promoter-binding manner [28]. It is probable that few amino acid changes of pORF2 result in binding activity differences.

The RNA secondary structure may also influence the stability and interaction of a virus genome inside host cells. The HEX1 protein of *F. graminearum*, the major component of peroxisome-derived Woronin body (WB) that seals the septal pore in response to hyphal wounding, has been found to be up-regulated in FgV1-DK21-infected *F. graminearum* [29]. It has been demonstrated that HEX1 can bind with 5′- and 3′-untranslated regions of plus-strand FgV1-DK21 RNA to form complexes [30]. The 5′-UTR sequence of FgV1-ch was the same as FgV1-DK21, while the 3′-UTR sequence was different. As shown in Figure 5, 3′-UTR of FgV1-ch and FgV1-DK21 were predicted with different loop structures. The binding ability of FgV1-ch with HEX1 might be influenced by these nucleotide changes, which required further demonstration. Other factors, such as host adaption, genetic diversity, and disease resistance level, may result in symptom differences between FgV1-ch and FgV1-DK21. 

## Figures and Tables

**Figure 1 viruses-12-00357-f001:**
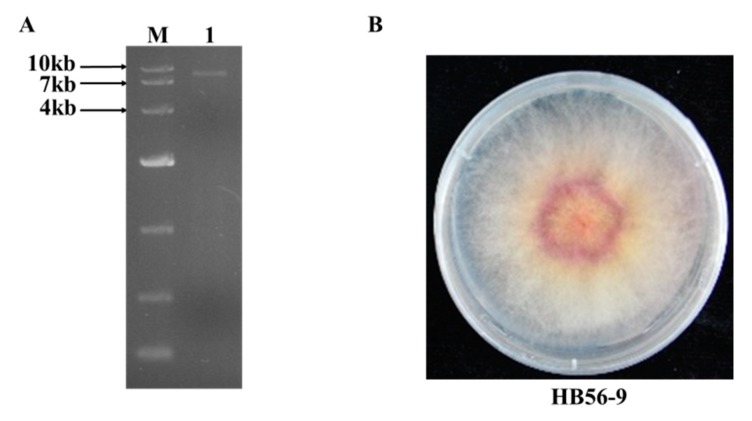
(**A**) dsRNA fraction extracted from *F. graminearum* strain HB56-9 was electrophoresed in a 1% agarose gel and visualized under UV light after staining with ethidium bromide. Lane M, DNA marker; Lane 1, dsRNA sample after treatment with S1 nuclease and RNase-free DNase I. (**B**) Colony morphology of strain HB56-9 after 4 days on PDA plate in the dark.

**Figure 2 viruses-12-00357-f002:**
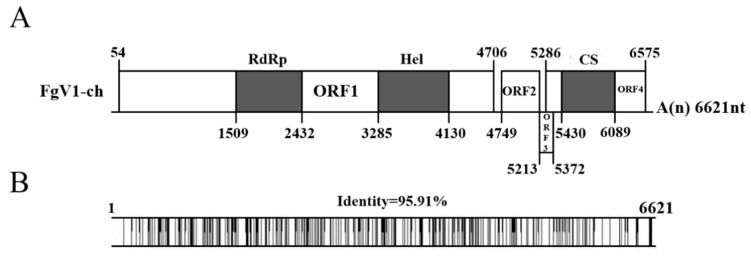
(**A**) Genome organization of FgV1-ch. FgV1-ch have four putative open reading frames (ORFs) that were represented by boxes. The numbers and solid lines above or below refer to the positions of initiation and termination codons of the respective ORFs. ‘A(n)’ represents poly (A) structure. The relative positions of the RdRp, Hel and CS domains in the proteins encoded by ORF1 and ORF4 are shaded in gray; (**B**) Full genome nucleotide alignment between FgV1-ch and FgV1-DK21. The different nucleotides along the genome are represented with vertical lines. The “1” and “6621” on the left and right indicate the start and stop nucleic acid position of FgV1-ch genome respectively.

**Figure 3 viruses-12-00357-f003:**
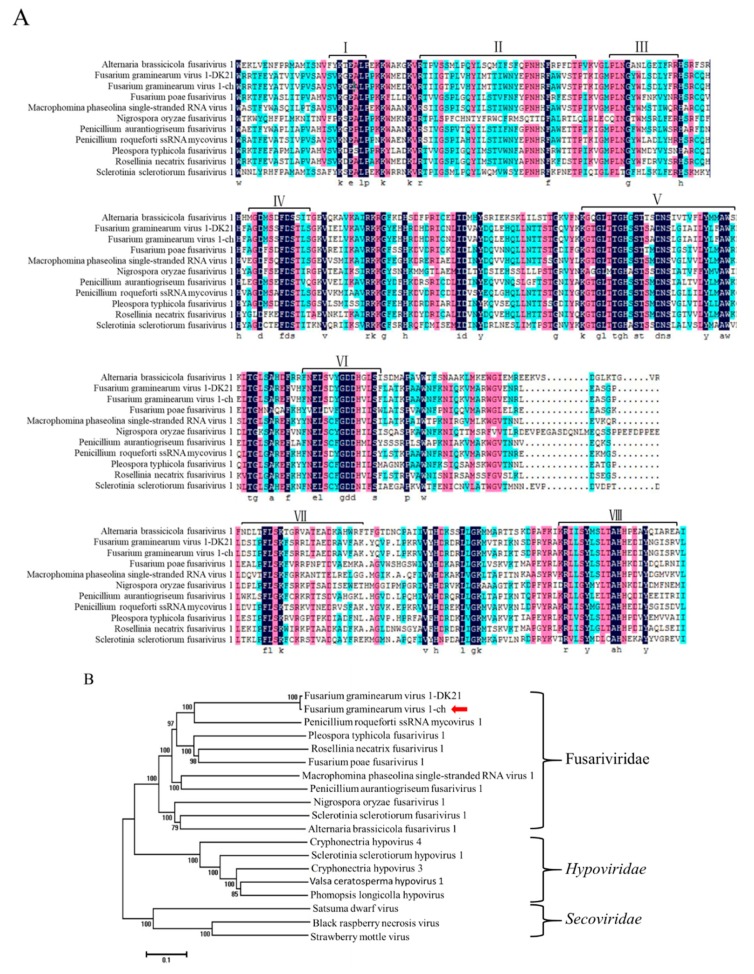
(**A**) Amino acid sequence alignment of FgV1-ch with those of other mycoviruses. Numbers I-VIII indicate the eight conserved motifs in RdRps. (**B**) Phylogenetic analysis of FgV1-ch (marked by arrow) and related RNA viruses. The phylogenetic tree was generated by NJ method in the MEGA 6.0 program. The numbers indicate the percentage of bootstrap replicates that support each branch node (only values >50% are shown). The scale bar at the lower left represents a genetic distance. The GenBank accession numbers are as follows: Fusarium graminearum virus 1 -DK21 (YP_223920.2); Penicillium roqueforti ssRNA mycovirus 1 (YP_009052456.1); Pleospora typhicola fusarivirus 1 (YP_009182158.1); Fusarium poae fusarivirus 1(YP_009272906.1); Rosellinia necatrix fusarivirus 1 (YP_009047147.1); Macrophomina phaseolina single-stranded RNA virus 1 (ALD89094.1); Penicillium aurantiogriseum fusarivirus 1 (YP_009182154.1); Sclerotinia sclerotiorum fusarivirus 1 (YP_009143301.1); Alternaria brassicicola fusarivirus 1 (YP_009222009.1); Nigrospora oryzae fusarivirus 1 (APA05125.1); Cryphonectria hypovirus 4 (YP_138519.1); Valsa ceratosperma hypovirus 1 (YP_005476604.1); Phomopsis longicolla hypovirus (YP_009051683.1); Sclerotinia sclerotiorum hypovirus 1 (YP_004782527.1); Cryphonectria hypovirus 3 (AAF13603.1); Black raspberry necrosis virus (CCE57809.1); Satsuma dwarf virus (NP_620566.1); Strawberry mottle virus (NP_599086.1).

**Figure 4 viruses-12-00357-f004:**
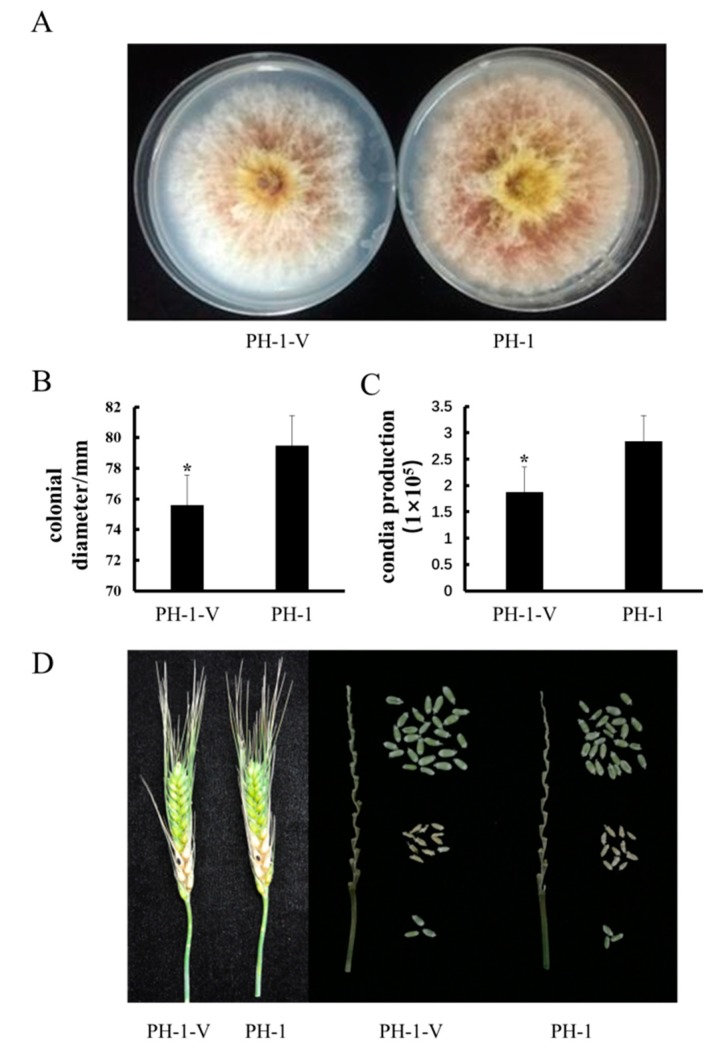
(**A**) Colony morphology of strain PH-1-V (virus-carrying) and PH-1 (virus-free) after 4 days of culture on PDA in the dark. (**B**) Colony diameter of strains PH-1-V and PH-1after 4 days of culture on PDA in the dark. “*” indicates a significant difference (*p* < 0.05). (**C**) Conidial production of strains PH-1-V and PH-1 after 5 days in CMC medium. “*” indicates a significant difference (*p* < 0.05). (**D**) Fusarium head blight symptoms caused by PH-1-V and PH-1 fungi in the fields. On the left side, the disease symptoms infected by virus-carrying and virus-free conidia were photographed after 15 days of inoculation in the flowering stage. The black dots on the wheat kernels indicate the inoculation positions. On the right side, PH-1-V- and PH-1-infected kernels were picked from stalks. Spikelets in the middle which were infected with virus-carrying and virus-free *Fusarium* strains were shriveled. Pathogenicity tests were repeated three times and replicated with at least 15 wheat heads for each strain.

**Figure 5 viruses-12-00357-f005:**
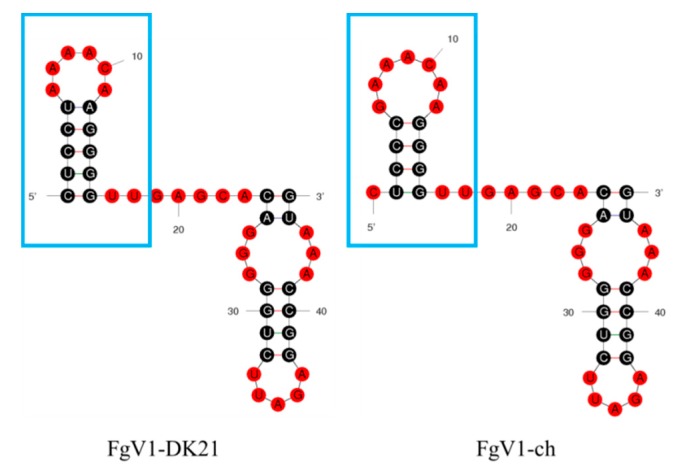
The predicted 3′-UTR secondary structures of FgV1-ch and FgV1-DK21. These secondary structures both contain two Hairpin loops (FgV1-DK21, Initial ΔG = −10.20 kcal/mol; FgV1-ch, Initial ΔG = −10.10 kcal/mol. Ionic conditions: 1M NaCl, no divalent ions.). The different loops are labelled with blue square boxes.

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
