# Peer review of "Molecular Characterization of a Novel Strain of Fusarium graminearum Virus 1 Infecting Fusarium graminearum"

_viruses, 2020, doi:10.3390/v12030357_

Round 1

Reviewer 1 Report

Zhang et al reported a new isolate of Fusarium graminearum virus 1, named FgV1-ch, that had mild effects on host mycelial growth, spore production, and virulence. FgV1-ch looks mostly the same as the previous isolate FgV1-DK21. It would be interesting to explore whether the different sites between two isolate are hypovirulent factors. Owning to host adaption mentioned in the discussion, it’s better to exclude the host genetic diversity that leads to different phenotypes.

What’s the e-value of CS domain?

Table 1, doesn’t look elegant.

The “F” in F. graminearum also should be in italic form.

The reference 2 in line 38 may be unsuitable.

The statistic analysis result should be marked in Fig 4B and C.

Line 43, “sensu stricto” is latin usage, and should be italic.

Line 48, “Extensive application of chemical fungicide has caused serious environmental pollution and fungi tolerance problems.” Missing reference.

Line 105, “ug/ml” should be Greek letter.

Line 168, delete “()” of Figure 2A. “-“ in “ORF1—4” should be English form.

Line 169, “mycoviruses” should be “FgV1-DK21”.

Line 172, add “-”between “RNA” and “dependent”,

In figure 2B, what’s the difference between full vertical line and short vertical line?

Line 191, delete “s” in “ORFs1-4” .

Line 192, line 193, delete “,” before “and”.

Line 224, delete “s” in “FgV1-chs”.

A ref is needed for discussion, line 284-287.

Line 384, a “.” missed after “cell death”.

Reviewer 2 Report

This paper reports molecular and biological characterization of a new strain ‘ch’ of Fusarium graminerarum virus 1 (FgV1). Since well-characterized FgV1 strain ‘DK21’ was previously shown to confer hypovirulence to its host Fusarium boothii, FgV1-ch is somewhat unique, in term of its trait conferring no significant hypovirulence to the host fungus Fusarium graminearum. The authors stated that FgV1-ch could be a useful counterpart in comparative analyses with FgV1-DK21, to understand molecular mechanism underlining phenomena occurred upon the infection of FgV1-DK21. Virus characterization was well performed in this study, but this reviewer felt that the paper leaves something to be desired because the study just understood the presence of a FgV1 strain having slight nucleotide changes with DK21 strain. More on comparative data may increase the impact of this study, for example, detailed comparison of amino acid changes in pORF2 (the most distantly related protein in FgV1 strains) may be worth. The pORF2 is a potential virulence determinant of FgV1-DK21 and -ch because of its uknown function as an RNA silencing suppressor (RSS). Each amino acid difference may be highlighted along with secondary structure prediction of pORF2 polypeptide. If the authors could address the RSS ability of pORF2 it is truly valuable, but it may be beyond the scope of this study. In any case, the authors should think more about how they can elevate the value of this study. Minor comments are as attached.

Reviewer 3 Report

This paper is about finding new strain of Fusarium graminearum virus 1, infecting F. graminearum. Followings are some points to improve the manuscript:

1-Title: 

Authors can find some changes in title, in attached file, to make it more clear.

2-Introduction

Line 38: Reference 2 is completely inappropriate here!

Line 40: Endornaviridae is also the family of ssRNA viruses that infect fungi.

Line 41: Do authors have any references for this sentence? Because as I know most discovered myxoviruses have dsRNA genomes.

Line 54: FgV2 also have hypovirulence effect on its host (Lee et al. 2014).

3-Materials and methods

Line 75: “The strain HB56-9 was obtained by single spore isolation and pure culturing.” Why was only one fungal strain pured? What about the other fungal strains?

Lines 100-101: “(Wang et al.,2013)” this reference is not cited correctly in text.

The authors must follow the guidelines of the journal keep constancy in citing the references.

Line 113: Reference 18 is not relevant to the context.

Lines 114-115: What is the concentration of conidia used for this experiment?

Lines 116-117: What is the statistical test used in this study?

Line 121: What kind of homology was searched? Which BLAST program was used?

Line 124: Please write the web server.

Lines 125-127: It is better to remove this section, and just mention about depositing the sequence in GenBank in results.

4-Results

Line 171: How many amino acids has the protein encodes by ORF1? 

Lines 191-194: If authors mention about sequence identities by detail in the context, there is no need for table 1. Table 1 is the repeat of these lines. Keep one of them, not both.

Lines 194-200: 11 sequences were used for alignment, but there are more viruses from different families in phylogenetic tree that are not in alignment. It is better authors correct this part.

5-Discussion

Lines 285-286: Reference?

Lines 297-298: The sentence is not clear. Needs more explanation.

6-Figures

Fig 4A. Unlike what is written in the text, the colony morphology is different between two strains, specially in pigmentation!

Fig4B-C: Is the difference between two strains statistically significant or not?

7-References

Reference #2 and 16: The author names are incomplete.

Reference #19: The paper name in incorrect.

8- It is better authors explain more in each section including materials and methods and results to present their study better. It is more desirable if authors make it more clear in the manuscript when they talk about nucleotide sequences and when talk about amino acid sequences.

9- Authors can find more comments in attached file.
